# Detection and Visualisation of Pneumoconiosis Using an Ensemble of Multi-Dimensional Deep Features Learned from Chest X-rays

**DOI:** 10.3390/ijerph191811193

**Published:** 2022-09-06

**Authors:** Liton Devnath, Zongwen Fan, Suhuai Luo, Peter Summons, Dadong Wang

**Affiliations:** 1School of Information and Physical Sciences, The University of Newcastle, Newcastle 2308, Australia; 2British Columbia Cancer Research Centre, Vancouver, BC V5Z 1L3, Canada; 3College of Computer Science and Technology, Huaqiao University, Xiamen 361021, China; 4Quantitative Imaging, CSIRO Data61, Marsfield 2122, Australia

**Keywords:** pneumoconiosis, coal worker pneumoconiosis, occupational lung diseases, computer-aided diagnosis, ensemble learning, machine learning classifiers, deep learning, CheXNet, chest X-ray radiographs, Grad-CAM

## Abstract

Pneumoconiosis is a group of occupational lung diseases induced by mineral dust inhalation and subsequent lung tissue reactions. It can eventually cause irreparable lung damage, as well as gradual and permanent physical impairments. It has affected millions of workers in hazardous industries throughout the world, and it is a leading cause of occupational death. It is difficult to diagnose early pneumoconiosis because of the low sensitivity of chest radiographs, the wide variation in interpretation between and among readers, and the scarcity of B-readers, which all add to the difficulty in diagnosing these occupational illnesses. In recent years, deep machine learning algorithms have been extremely successful at classifying and localising abnormality of medical images. In this study, we proposed an ensemble learning approach to improve pneumoconiosis detection in chest X-rays (CXRs) using nine machine learning classifiers and multi-dimensional deep features extracted using CheXNet-121 architecture. There were eight evaluation metrics utilised for each high-level feature set of the associated cross-validation datasets in order to compare the ensemble performance and state-of-the-art techniques from the literature that used the same cross-validation datasets. It is observed that integrated ensemble learning exhibits promising results (92.68% accuracy, 85.66% Matthews correlation coefficient (MCC), and 0.9302 area under the precision–recall (PR) curve), compared to individual CheXNet-121 and other state-of-the-art techniques. Finally, Grad-CAM was used to visualise the learned behaviour of individual dense blocks within CheXNet-121 and their ensembles into three-color channels of CXRs. We compared the Grad-CAM-indicated ROI to the ground-truth ROI using the intersection of the union (IOU) and average-precision (AP) values for each classifier and their ensemble. Through the visualisation of the Grad-CAM within the blue channel, the average IOU passed more than 90% of the pneumoconiosis detection in chest radiographs.

## 1. Introduction

Clinical expertise and observer variability are reduced by computer-aided diagnosis (CADx) systems, which have become increasingly popular in medical imaging. The last few de-convolutional neural networks (CNNs)-based data-driven deep learning (DL) algorithms have performed well for chest X-ray (CXR) screening. However, due to the low prevalence of some diseases and restrictions on sharing patient data, transfer learning of CNN models became a popular technique, in which pre-trained CNN models from one application domain were used to provide a foundation for new CNN models in a different application domain. Transfer learning improves a model’s performance if it has been trained on datasets of a related domain to those of the problem being solved [1]. Therefore, if the goal is to detect any disease from medical images, then a model that has already gained knowledge from a similar domain should be selected. This can reduce learning time and improve the performance with a small training dataset of a CNN in the new application domain [2]. To combat the issue of limited data, feature extraction could be performed via transfer learning of deep learning models, with the resulting features being fed into new classifiers. In many medical image classifications [3,4,5,6,7], deep learning models were used as a feature extractor, where the extracted features were fed to different machine learning classifiers, such as Naive Bayes [8], Multilayer Perceptron (MLP) [9], Support Vector Machines (SVM) [10], K-Nearest Neighbours (KNN) [11], Random Forest (RF) [12], and Decision Trees (DT) [13].

Deep learning models are critical for establishing trust and demonstrating their ability to integrate with computer-aided detection. Over the last year, the interpretation and visualisation of deep CNN features has grown in popularity for understanding the modelling and behaviour of the features learned across the trained model’s convolutional layers [14,15]. It was observed that the convolutional layers of CNN retain all spatial information depending on the increased variance of the input image. This spatial information may differ on the depth of the convolutional layer or block within CNN structure, which is lost in fully-connected layers. As a result, the final convolutional layers or blocks represent the best understanding of the information between all layers. The neurons in the layer search for pertinent information on the specific classes of the image. In [15], a visualisation strategy was presented called class-activation-map (CAM) to locate image regions of interest (ROIs), which are relevant to an image category. However, the use of this strategy is limited because it only works with deep learning architecture. Subsequently, a common CAM technique was proposed based on the gradient-weighted CNN input class activation mapping, referred to as Grad-CAM with universal tuning form [16], which the information based on the convolutional layer gradient uses to assign significance values to each neuron in the ROI. Medical image processing researchers also applied Grad-CAM to explain the disease predictions of the CNN model and interpreted the depictions learned with CXR [17,18,19,20,21,22]. The concept behind Grad-CAM is to compute the gradient of the ranking score in relation to the CNN characteristics map. It highlights the specific ROIs based on the greatest gradient score.

In this paper, we have proposed an ensemble technique of multidimensional features learned from CNN to detect and visualise pneumoconiosis disease in CXRs. Our list of contributions is summarised below:I.We have utilised posterior–anterior (PA) CXRs databases compiled by Coal Services Health NSW, St. Vincent’s Hospital Sydney, Wesley Medical Imaging Queensland, and the International Labour Organization (ILO).II.We have applied an efficient CNN architecture, CheXNet-121 to learn and extract multidimensional features from three different folds of database.III.Individually for each fold, four different sets of multidimensional features (F1024, F512, F256, and F128) were extracted by using the supreme dense block functionality of the CheXNet-121 architecture.IV.In order to detect pneumoconiosis, extracted features were used as input for nine traditional machine learning classifiers, such as SVM-SF (SVM with SF kernel), SVM-RBF (SVM with RBF kernel), SVM-PF (SVM with PF kernel), Gaussian Naive Bayes (GNB), Multi-layer Perceptron (MLP), Radius Neighbours (NB), K-Nearest neighbours (KNN), Decision Trees (DTs), and Extra Decision Trees (EDTs)V.To determine the optimal prediction label, an ensemble of nine decisions was made using the majority voting (MVOT) system.VI.Ensemble learning performance on each dataset associated with four feature sets was assessed using eight metrics: TPR (true positive rate or recall or sensitivity), TNR (true negative rate or specificity), precision, FPR (false positive rate), FNR (false negative rate), F1-Score, ACC (accuracy), and MCC (Matthews correlation coefficient).VII.To compare the efficiency of the classifiers, AUC-PR (area under the precision–recall curve) and AP (average precision) values were calculated.VIII.Grad-CAM was applied to produce a coarse localised map highlighting the most important ROIs in the lung to predict the disease of coal workersIX.Additionally, the Grad-CAM localisation outputs on the RGB image were split into red, green, and blue channels, where each colour specified clearer discriminative ROIs.X.Finally, we have compared the ROI highlighted by Grad-CAM with the ground-truth ROI based on the intersection of the union (IOU) and average-precision (AP) values of each classifier and their ensemble.

## 2. Research Background

This section provides context for this study, including previous studies and findings for pneumoconiosis classification on the same dataset using various classical, traditional machine, and deep learning methods. Furthermore, Section 2.2 summarises the various machine learning classifiers used in this study.

### 2.1. Previous Study

Radiology shows an increase or decrease in lung density on chest X-rays. Pulmonary opacities are dense abnormalities on chest X-rays. Pulmonary opacities include consolidation, interstitial, and atelectasis. Interstitial pulmonary opacities cause pneumoconiosis [23]. The international labour organisation (ILO) divides pneumoconiosis into two types: parenchymal and pleural. Small opacities (round or irregular) of 1.5 mm diameter (round) or 1.5 mm width (irregular) or less than or equal to 50 mm in size indicate parenchymal abnormalities [24]. An abnormal CXR wall shows angle obliteration and thickness diffusion in pleural abnormalities [25]. Previously, pneumoconiosis was classified using classical, traditional machine learning, and deep learning methods.

In classical methods, the abundance of small round opacities and ILO extent properties indicated normal and abnormal lungs [26,27,28,29,30,31,32,33,34,35]. Backpropagation neural networks were used to determine the shape and size of round opacities in region of interest (ROI) images [36,37,38], and the abnormalities were classified and compared to the standard ILO opacity measurement. Using the same data set that was used in this study, this method worked 83.0% of the time to find pneumoconiosis [39].

Handcrafted feature extraction or selection were used in traditional machine learning for pneumoconiosis detection. Handcrafted features like texture [30,32] were extracted from the left–right lung zones [40,41,42,43]. Following feature selection, they were fed into various machine learning classifiers, including support vector machines (SVM) [40,41,43,44,45,46,47,48,49,50,51], decision trees (DT) [47,48], random trees (RT) [44,49,50,51], artificial neural networks (ANNs) [52,53,54], K-nearest neighbours (KNN) [55], self-organizing map (SOM) [55], backpropagation (BP), the radial basis function (RBF) neural networks (NN) [44,49,50,51,55,56], and Ensemble classifier [41,43,48]. Among the classifiers, SVM had the best overall detection accuracy, with a 73.17 percent success rate when using the same dataset as this study.

Recent advances in deep learning have been made possible by high-dimensional data representation [57,58]. In medical image processing [59], deep CNNs outperform humans in detecting cancer markers in blood and skin [60,61,62], malaria in blood cells [63], and respiratory diseases in chest X-rays [39,64,65,66,67,68,69,70,71,72].

On the same dataset, we have used different deep learning approaches. We first implemented convolutional neural networks (CNN) with and without transfer learning, using the base models, DenseNet-121 [19] and CheXNet-121 [18]. Furthermore, with an overall accuracy of 90.24 percent, we were able to develop a cascade model that outperformed others in detecting pneumoconiosis. Prior research using the same dataset revealed that the pre-trained deep learning model, CheXNet-121, outperformed classical and traditional approaches.

### 2.2. Machine Learning Classifiers

In machine learning, support vector machine (SVM) networks can be used as either a linear or a non-linear classifier. In the non-linear case, SVM can be utilized to address binary classification problems with different kernel functions which maximize the marginal distances between two classes. Sigmoid, radial basis, and polynomial are three popular kernel functions for SVM. One of the activation functions of SVM-SF (sigmoid function) is hyperbolic tangent function; SVM-RBF (radial basis function) uses exponential functions as its activation function, which measures the Euclidean distance between the classes; SVM-PF (polynomial function) reduces the similarity of vectors that optimize the network’s learning capacities [73,74].

In supervised learning, a simple classification algorithm called Gaussian Naïve-Bayes (GNB), which is based on Bayes probability theorem, can be trained to work for small sets of features vectors [74]. Each output feature vector from a multi-layer perceptron (MLP) is an input to the following layer, and a non-linear activation function forecasts the probabilities of unknown vectors [73]. MLP can classify the datasets which are not linearly separable. Both the radius neighbours (RN) and the k-nearest neighbours (KNN) classifiers operate in a specific vector space where each feature vector maintains a constant separation from a random vector point. The term “radius of neighbours” refers to the fixed distance. The KNN displays the vectors that would connect the top k nearest neighbours’ features in the same vector space [75]. In a complex variable problem, classification becomes more difficult due to the complexity of feature patterns. In this case, a model randomly selects some target feature sets that are called decision trees (DTs), where the leaves and branches of DTs represent the class labels and features of these labels. For the complicity of features a single DT sometimes over-fits the training dataset, while the extra decision trees (EDTs) reduce the overfitting of the training model that can optimize the prediction probabilities [76]. A wide range of real-world applications, including medical image analysis, have shown that machine learning classifiers are extremely effective in a variety of situations [77,78,79,80]. In recent years, due to the low prevalence of some diseases and restrictions on the sharing of patient data, the machine learning classifiers summarised above have merged CNN features to great effect as well [3,4,5,6,7].

## 3. Datasets and Methods

The first part of this section discusses our dataset and how it was processed using randomised cross-validation to perform proposed ensemble technique. In contrast, the rest of the section describes the details of the detection and visualisation techniques implemented in this research.

### 3.1. Datasets

We have made use of the posterior–anterior (PA) CXRs databases that were accumulated by the Coal Services Health NSW, St. Vincent’s Hospital Sydney, Wesley Medical Imaging Queensland, and ILO. The database contained total 153 images, including 71 pneumoconiosis CXRs. To keep the training data balanced, 112 X-rays (56 normal and 56 pneumoconiosis) were used for training and 41 X-rays (26 normal and 15 pneumoconiosis) were used for testing. 25% of the training data was kept as a validation data set in order to select the best model weights based on validation performance. We repeated the randomised selection three times before dividing our total dataset into three folds, namely randomised 3 cross-fold Dataset 1, Dataset 2, and Dataset 3, as shown in Figure 1.

### 3.2. Method

Gao Huang et al. developed DenseNet-121, a CNN with dense connections between layers that was trained on the ImageNet database of 1000 classes, in 2017 [19]. The four dense blocks were utilised to make these connections, which consisted of joining them in such a way that their output sizes were similar. Rajpurkar, P. et al. transferred DenseNet-121’s knowledge from the ImageNet domain to the Chest X-ray domain and released hybrid CheXNet-121 pre-trained model [18]. This CheXNet-121 model was trained on ChestX-ray14, which contains 112,120 frontal X-ray pictures from 30,805 individuals [81]. This study proposes a subsequent implementation of transfer learning to apply CheXNet-121’s knowledge to a tiny dataset of X-ray diseases that includes none of those 14 classes.

The CheXNet-121 architecture represents a discriminating level of characteristics after each convolutional block that is more robust with bigger datasets. The CheXNet-121 has four convolutional dense blocks followed three transition layers which are fully connected, as demonstrated in Figure 2. Each of the four dense blocks was made up of 6, 12, 24, and 16 times of the 1 × 1 convolution and 3 × 3 convolution, with features being multidimensional, as directed by arrows in Figure 2. On the input feature maps, each layer adds a few new features in a dense block, which causes the feature size to increase. The transition layers can do downsampling by using a batch-norm layer, a 1 × 1 convolution operation, and then a 2 × 2 average pooling outside dense blocks, as shown in Figure 2. This makes sure that the size of the feature map inside dense blocks stays the same so that features can be concatenated. The initial dense blocks’ features are regarded as low-level in comparison to the fourth dense blocks.

In this section, we have proposed an ensemble learning using nine machine learning algorithms to classify multidimensional high-level CheXNet-121 features learned from chest X-rays. The CheXNet-121 model was used as a feature extractor by removing the last layer close to the output layer. Next, a global average pooling layer was added which converted the output of the model into one-dimensional vectors. The resolution of input X-ray images is 512×512 pixels with three channels. Before use of the model, we compiled it using an Adam optimizer with low learning rate (0.0001) and binary cross entropy as a loss function. The main contribution of this section is summarized as follows:

Firstly, we extracted four sets of multidimensional high-level CheXNet-121 features (F1024, F512, F256 and F128) independently from the three randomized cross-fold datasets as discussed in Figure 1. Here, each feature set indicates the number of features extracted from each image. Secondly, extracted features were used as the input of nine classifiers, including SVM-SF (support vector machine with sigmoid function kernel), SVM-RBF (support vector machine with radial basis function kernel), SVM-PF (SVM with polynomial function kernel), Gaussian Naive Bayes (GNB), multi-layer perceptron (MLP), radius neighbours (NB), k-nearest neighbours (KNN), decision trees (DTs), and extra decision tree (EDTs), as clearly mentioned in Figure 2. Thirdly, the optimal prediction label was calculated using the strategies of a majority voting (MVOT) system. Finally, the ensemble learning performances among three randomized cross-fold datasets were computed using eight metrics: TPR (true positive rate or recall or sensitivity), TNR (true negative rate or specificity), precision, FPR (false positive rate), FNR (false negative rate), F1-Score, ACC (accuracy), and MCC (Matthews correlation coefficient). Here, the MCC provides a more accurate statistical measure based on the four confusion matrix values of true positives, false negatives, true negatives, and false positives. Therefore, a model will achieve a higher MCC score if and only if it achieves a good return in the four matrix values.

The hyper parameters of all classification algorithms were chosen based on experiments aiming to achieve the best performance of each model on validation features. For SVM, we tested different values for each parameter; for example, five values (0.00001, 0.0001, 0.001, 0.01, and 0.1) are assessed for the penalty C, and the parameter gamma, γ was assigned a set of values as $2^{-9},\;2^{-8},\;2^{-7},\;2^{-6},\;2^{-5},\;2^{-4},\;2^{-3},\;2^{-2},\;2^{-1},\;2^{0},\;2^{1},\;2^{2},\;2^{3},\;2^{4},\;2^{5},\;2^{6},\;2^{7},\;2^{8},\;2^{9}$. For GNB we set only the variance smoothing parameters $10^{-10},\;10^{-9},\;10^{-8},\;10^{-7},\;10^{-6},\;10^{-5},\;10^{-4},\;10^{-3},\;10^{-2},\;10^{-1}$ which were stable despite the features variances. For MLP, we used the default hidden layer size 100, rectifier linear units (relu) as its activation function, L2 penalty weights regularization, Adam optimizer, and initial learning rates (0.00001, 0.0001, 0.001, 0.01, and 0.1) with maximum iteration 10,000. The range of k neighbours was tuned odd number from (1 to 84) in KNN, and fractional radius neighbour parameters from (1.5 to 5.00), which may vary depending on the training feature sets used in RN. For the maximum depth of trees, it was tuned with odd integer numbers from 1 to 30. The other parameters of all classifiers were chosen as default values of the algorithms. The best tuned parameters were used for training and testing, which gave optimal performance on the extracted feature sets, F1024, F512, F256, and F128.

The recall and precision values for each test CXR utilising the feature set of its associated dataset were presented in Figure 3, Figure 4, Figure 5, Figure 6, Figure 7, Figure 8, Figure 9, Figure 10, Figure 11, Figure 12, Figure 13 and Figure 14 to demonstrate how well a single classifier performed in an ensemble. As a result, using their AUC-PR (area under the precision-recall curve) and AP (average precision) values made comparing efficiency easier. Trapezoidal numerical integration of the precision–recall curve yields the AUC value. The weighted average of the precision obtained at each threshold, with the recall increase being the preceding threshold used as weight, is how AP summarises a graph. This implementation is not interpolated, as opposed to the PR curve AUC calculation, which uses linear interpolation and can be overly optimistic.

Additionally, we applied the Grad-CAM (gradient-weighted class activation mapping) technique to interpret the most important ROIs that contributed to the classification of pneumoconiosis disease. Finally, we have interpreted the localisation performance of Grad-CAM through four convolutional blocks within the CheXNet-121 architecture, as demonstrated in Figure 2. Next, we split the Grad-CAM image of the high-level feature block into red, green, and blue, as shown in Figure 15. We compared all predicted ROIs with ground truth ROIs based on the measurement of the intersecting area over the union (IOU) between them and best AP values of each classifier and their ensemble. IOU is used to measure the performance of object detection from the overlapping relationship area between the ground-truth and predicted bounding box provided for a particular input image [82,83]. The details proposed Grad-CAM technique of pneumoconiosis detection is presented in Figure 15.

We implemented our proposed model using the Keras version of CheXNet-121 with scikit-learn 0.19.1 machine learning classifiers, matplotlib 3.1.3 libraries, and Python 3. The experiments were conducted on a high-performance computing system at the University of Newcastle.

## 4. Results and Discussions

The ensemble outcomes for the nine traditional machine learning classifiers—SVM-SF, SVM-RBF, SVM-PF, GNB, MLP, KNN, RN, DTs, and EDTs—based on the majority voting (MVOT) system are presented in this section. Table 1 displays the results of our investigation into the ensemble learning performance of nine classifiers using eight evaluations based on the MVOT confusion matrix. For the randomised cross-fold Datasets 1 and 2, the ensemble classifiers attained an accuracy of 90.24 percent for the feature sets F1024 and F512, as discussed in Figure 1.

For the features sets F256 and F128, the ensemble classifiers achieved an accuracy of 92.68% and 90.24% for same Dataset 1 and 2. For features sets F1024 and F512, the ensemble classifiers achieved an accuracy of 87.80% for Dataset 3. For the features sets F256 and F128, the ensemble classifiers achieved an accuracy of 85.37% for Dataset 3. Therefore, average and maximum accuracies of 90.24% and 92.68% were achieved among the three datasets by the proposed ensemble learning method. Furthermore, we found that the highest MCC score, 85.86 percent, was attained on our imbalanced testing dataset. A low false positive rate (FPR) and false negative rate FNR was also achieved by the ensemble of nine classifiers. For the purpose of pneumoconiosis disease detection, the proposed integrated method also fares better than the alternatives on measures such as precision, recall, specificity, and F1-score.

In our proposed datasets, we used a balanced distribution of two classes during training and validation, but the test was imbalanced. Therefore, all test datasets contained 36.58 percent (15 out of 41) of pneumoconiosis and 63.42 percent (26 out of 41) of normal images. The precision–recall (PR) values for each test data set were plotted with the corresponding feature sets F1014, F512, F256, and F128, respectively. Then, the values AUC-PR (area under the PR curve) and AP (average accuracy) were computed to provide a different perspective on the evaluation of the result of the binary classifier. The Lager AUC-PR value shows better model performance in order for the PR curve to move to the ideal classifier. Varying according to the imbalanced data, the baseline of the PR curve is the horizontal line parallel to the x-axis value of the positive rate that would be the lowest precision value. Almost all real-world examples will be somewhere between ideal and baseline, which is not perfect but provides better forecasts than the ‘baseline’.

We demonstrate the AUC-PR and AP values for each classifier that has used three different datasets in the ensemble learning process using four dimensions of CheXNet-121 features in the following figures. All of the PR curves have demonstrated the equilibrium between the positive prediction values and the true positive rate through the application of a variety of probability thresholds. The average precision (AP) is a representation of the mean of the precision values obtained at each recall of a new positive sample. This indicates whether a classifier is capable of accurately identifying all instances of pneumoconiosis without mistakenly labelling an excessive number of normal occurrences as pneumoconiosis. As a consequence of this, the AP is high when the classifier is capable of accurately detecting pneumoconiosis disease in chest X-ray radiographs.

Figure 3, Figure 4, Figure 5 and Figure 6 show the PR curves in an ensemble of nine classifier performances using the corresponding CheXNet-121 feature sets, F1024, F512, F256, and F128 of Dataset 1. The highest AUC-PR and AP values on F1024 were 0.8966 and 0.9019, respectively, achieved by the same SVM-SF classifier. The highest AUC-PR and AP in the RP curves on F512 were 0.9007 and 0.9046, respectively, achieved by the two classifiers, SVM-SF and MLP. The highest AUC-PR and AP pairs obtained by the same RN classifier in the PR curves of F256 and F128, respectively, are (0.9507 and 0.9490) and (0.9302 and 0.9291).

Figure 7, Figure 8, Figure 9 and Figure 10 show the PR curves in an ensemble of nine classifier performances using CheXNet-121 feature sets F1024, F512, F256, and F128 from Dataset 2. The highest AUC-PR and AP pairs in the PR curves of F1024 and F512 are (0.9208 and 0.9234) and (0.8835 and 0.8876), performed by the same MLP classifier. The pairs of AUC-PR and AP (0.9601 and 0.8655) and AP (0.9064 and 0.9088), on the other hand, were produced as the highest on F256 and F128 with two isolated classifiers, SVMRBF and GNB.

Figure 11, Figure 12, Figure 13 and Figure 14 show the PR curves in an ensemble of nine classifier performances using feature sets F1024, F512, F256, and F128 from Dataset 3. With the help of the set of high-dimension extracted CheXNet-121 features, F1024, KNN, and MLP achieved the highest AUC-PR and AP values of 0.8385 and 0.7920 out of nine classifiers. By obtaining the highest AUC-PR and AP, the SVM variant, SVM-RBF, demonstrated effectiveness in the detection of pneumoconiosis using both F512 and F256. MLP obtained the lowest AUC-PR and AP values for F128 Dataset 3.

The ensemble technique showed that a set of nine classifiers for Datasets 1–3 performed average precision and recall values with F128 and F1024 testing sets. In most classifiers, positive predictive values increased, consistent with higher true positive rates. Most of the PR curve is oriented toward the ideal classifier and enclosed by the baseline. Although the proposed ensemble methods have increased the accuracies of Dataset 2 from 87.80% to 90.24%, and Dataset 3 from 85.37% to 87.80%, they did not improve the accuracy of Dataset 1 from 92.68%. The optimum pneumoconiosis detection accuracy and MCC scores of 92.68% and 85.86% were obtained with the F128 features. Thus, the higher MCC score indicated that the ensemble method obtained a good prediction across categories of the confusion matrix, true positives, false negatives, true negatives, and false positives, respectively. Maximum AP (average precision) values of 0.9490, 0.9234, and 0.9005 for Datasets 1–3 indicate RN, MLP, and SVMRBF were more accurate at detecting pneumoconiosis in chest X-rays.

Grad-CAM was used to calculate the automatic differentiation of the selected class final score with respect to the weights in each CheXNet-121 block’s feature map. The Grad-CAM is created by averaging multiple CAMs (class activation mapping) generated by the CheXNet-121 model’s four convolutional blocks. Figure 15 depicts the workflow.

We avoided CAMs before the final block in this study because the deepest convolution layer of each CNN model generates efficient characteristic maps. To demonstrate the efficacy of our technique, we separated three colour channels from the proposed Grad-RGB CAM’s outputs using the CheXNet-121 model and quantitatively compared their visual ROI localisation performance in terms of IOU and AP values. IOU evaluates the accuracy of the Grad-CAM highlighted region of interest (ROI) by splitting and combining the test image’s red, green, and blue channels, as shown in Figure 15.

Grad-CAM demonstrated that the ROI channels’ variance in intensity reflects the relative importance of each channel to the class. The ground-truth (green-bounding) and predicted (yellow-bounding) boxes represented the precise coordinates of the suspicious and Grad-CAM highlighted ROIs within the pneumoconiosis CXRs, respectively, and were used to calculate the IOU.

In Figure 15, we visualised the Grad-CAM outputs of each convolutional block within the CheXNet-121 architecture. In Section 3.2, the chemistry of these blocks is described in detail using Figure 2. The initial two convolutional blocks highlighted the overall pixels, size and shape of pneumoconiosis using Grad-CAM. The third block highlighted several suspected areas in the pneumoconiosis images. Our objective was to predict the most conspicuous suspect ROIs that matched with the most ground-truth ROIs, which we noticed from the high-level feature block’s outputs. Therefore, we avoided the first three blocks’ outputs in the measurements of intersection over union (IOU). The formula of IOU calculation has been attached in Figure 15.

As we know that the lungs of pneumoconiosis patients look black instead of a healthy pink, we turned our IOU calculation into three channels of input red-green-blue (RGB) images, as in Figure 15. Consequently, we observed the IOU-RGB differences in the accuracy of each channel. In Figure 15 and Figure 16, we showed a total of six Grad-CAM applications using two images from each of the three different test sets, where IOU accuracy has been demonstrated at each RGB and its separate channels independently. The comparison of IOU-RGB with each channel focused different accuracy among them, where only the IOU-BLUE has shown higher accuracy than IOU-RGB with the same bounding boxes.

Table 2 demonstrates the comparison between average-IOU and average positive prediction value (PPV or precision) through three cross-fold datasets. The average IOU passed an average of more than 85% of the pneumoconiosis detection in chest radiographs through the visualisation of the Grad-CAM datasets, where the average maximum was 89.32% for test Dataset 1.

Ensemble learning with the MVOT and the proposed method gave AP accuracy of 100% on test Dataset 3, but only 92.31 percent on the other test datasets. We also made a list of the highest AP that nine classifiers got from the precision–recall curve in Table 2. The RN classifier in Dataset 1 got the highest AP, which was 94.90 percent. SVMRBF used three-fold datasets of pneumoconiosis disease to figure out the overall 90 percent AP. The proposed ensemble learning achieved 92.68 percent pneumoconiosis detection accuracy in chest X-ray radiographs, as shown in Table 1.

In this research, we have presented a deep learning-based method with ensemble learning. The experimental results show that it performed well for the detection of pneumoconiosis and achieved an accuracy of 92.68%. An average precision of 92.31%, 92.31%, and 100.00% were achieved by the proposed technique. The proposed CheXNet-121 model visualisation helped to interpret the model’s behaviour, compensated for the error of missing ROIs using individual CNN models, and demonstrated superior ROI detection and localisation performance compared to any individual constituent model.

## 5. Conclusions

We conducted a study to identify pneumoconiosis disease in CXRs using the deep learning technique. We explored deep learning applications through ensemble learning. The experimental findings show that the proposed methodology is an encouraging and better approach than applying a single model individually and other state-of-the-art approaches. In addition, the Grad-CAM visualisation helped interpret the behaviour of the model and demonstrated discrimination in detecting and localising ROIs within each RGB channel when compared to any individual convolutional block in the deep learning model. The experimental results show that the proposed method can be used for pre-screening of chest X-rays for the effective detection of pneumoconiosis. Expert radiologists can then spend more time on those X-rays that were flagged as having pneumoconiosis by the proposed model. We think the proposed frameworks are useful for the development of robust models in the classification of medical images and localisation of the superior ROI.

## Figures and Tables

**Figure 1 ijerph-19-11193-f001:**
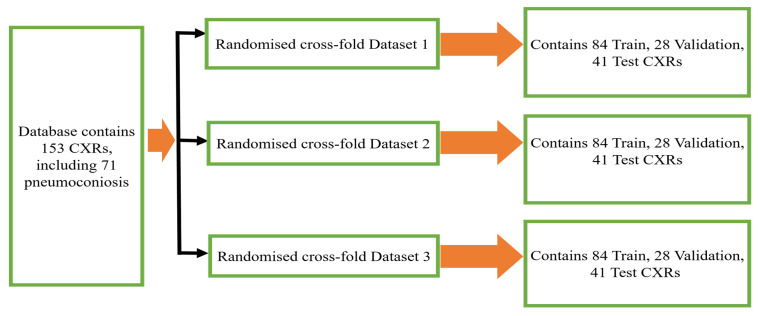
Three randomized cross-fold datasets of original database.

**Figure 2 ijerph-19-11193-f002:**
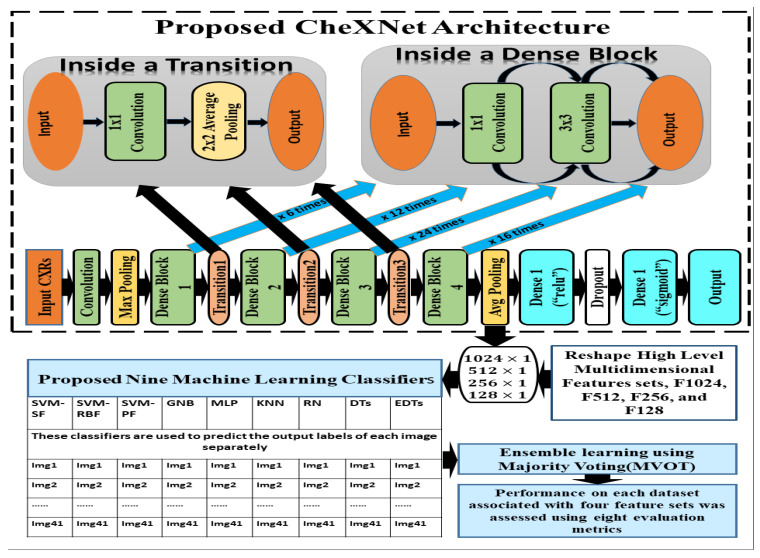
The details of our proposed ensemble learning of pneumoconiosis detection.

**Figure 3 ijerph-19-11193-f003:**
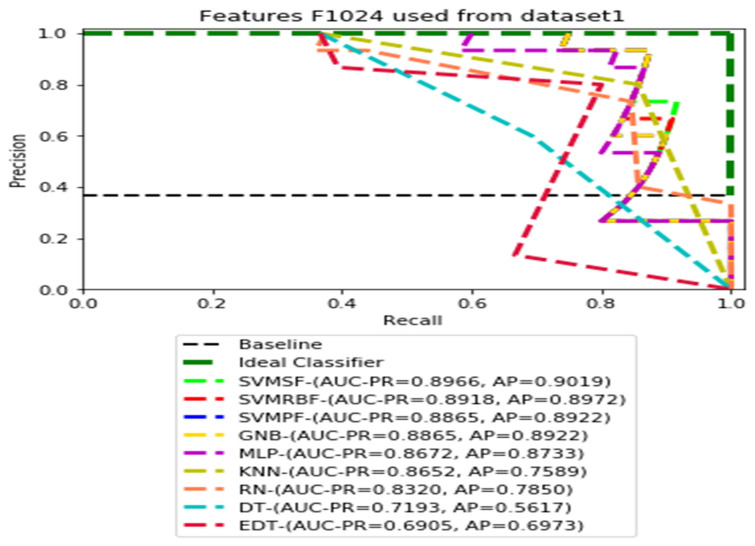
PR curves using feature set F1024 as the input of nine classifiers with Dataset1.

**Figure 4 ijerph-19-11193-f004:**
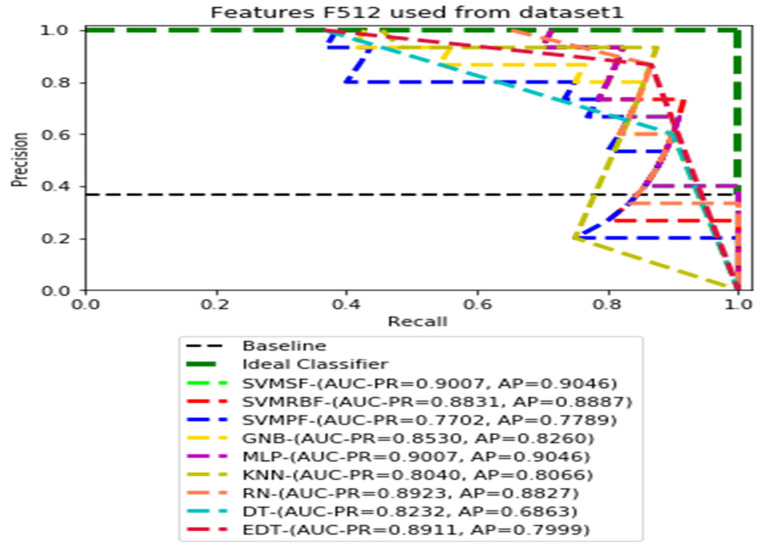
PR curves using feature set F512 as the input of nine classifiers with Dataset 1.

**Figure 5 ijerph-19-11193-f005:**
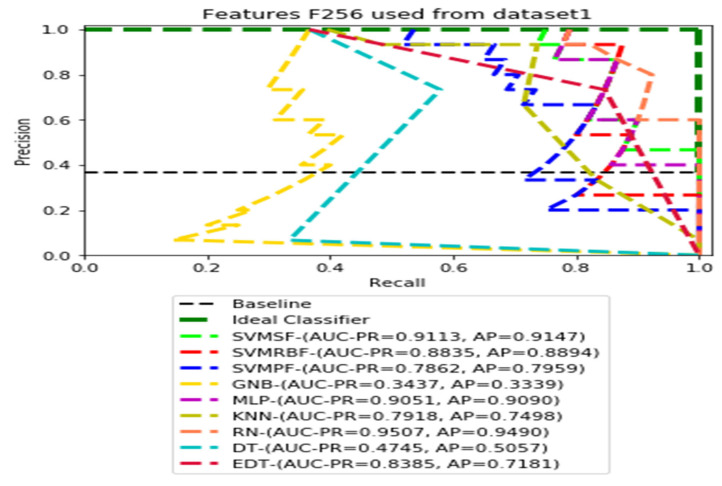
PR curves using feature set F256 as the input of nine classifiers with the Dataset 1.

**Figure 6 ijerph-19-11193-f006:**
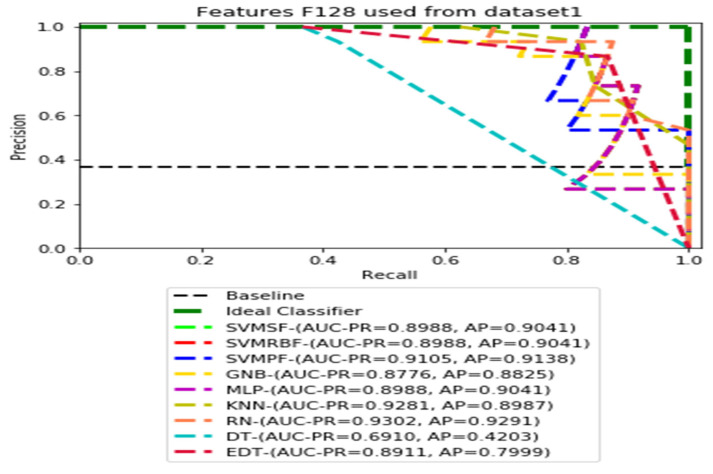
PR curves using feature set F128 as the input of nine classifiers with the Dataset 1.

**Figure 7 ijerph-19-11193-f007:**
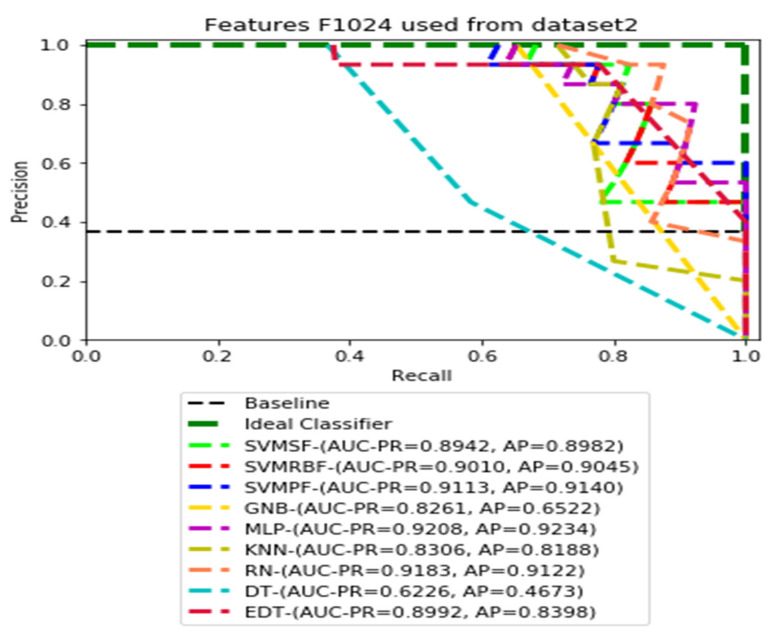
PR curves using feature set F1024 as the input of nine classifiers with Dataset 2.

**Figure 8 ijerph-19-11193-f008:**
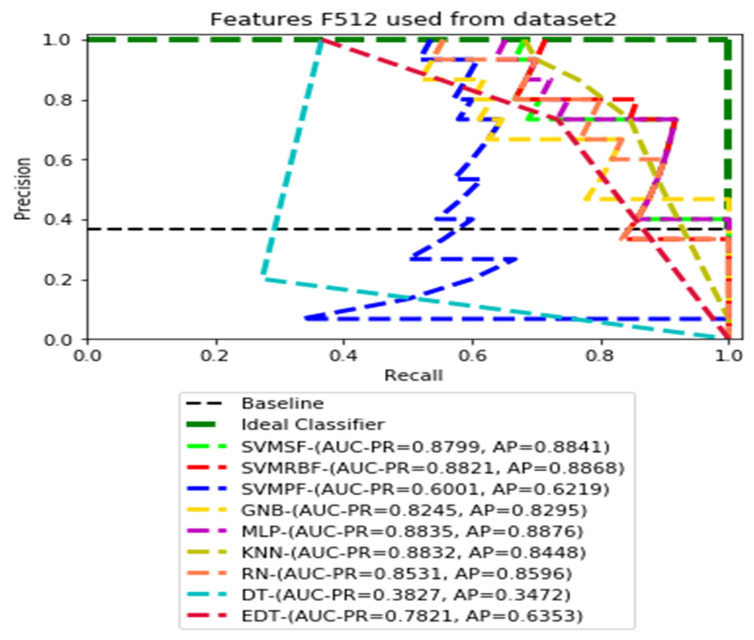
PR curves using feature set F512 as the input of nine classifiers with Dataset 2.

**Figure 9 ijerph-19-11193-f009:**
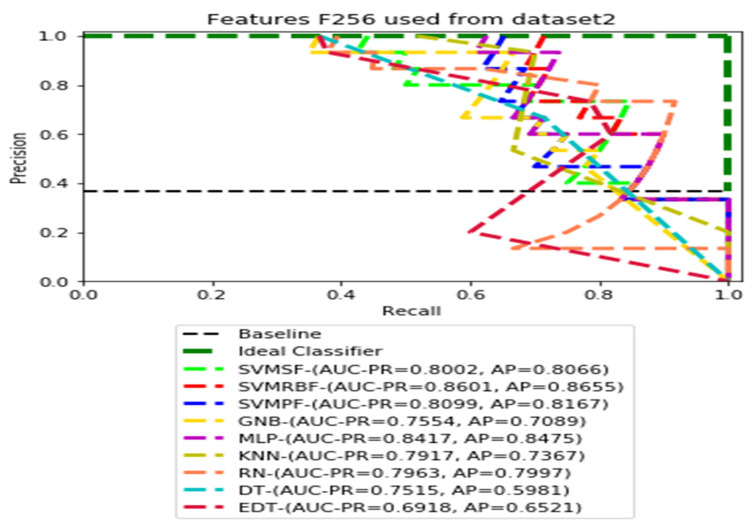
PR curves using feature set F256 as the input of nine classifiers with Dataset 2.

**Figure 10 ijerph-19-11193-f010:**
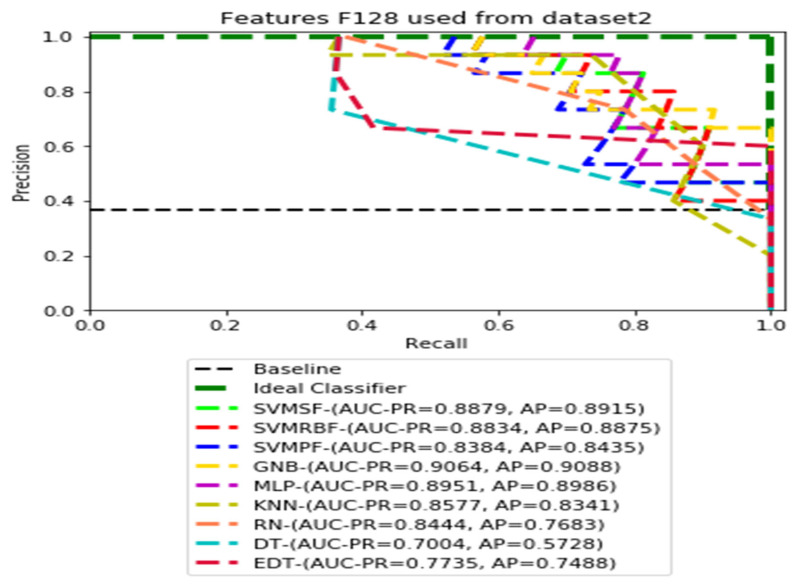
PR curves using feature set F128 as the input of nine classifiers with Dataset 2.

**Figure 11 ijerph-19-11193-f011:**
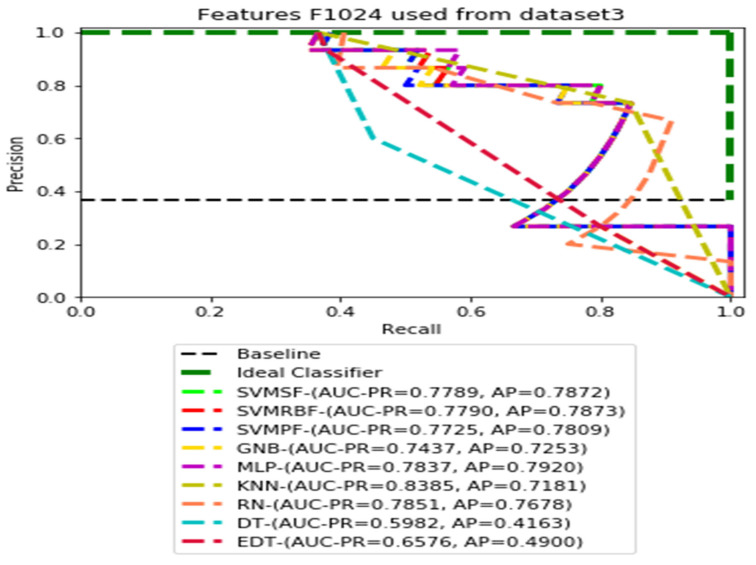
PR curves using feature set F1024 as the input of nine classifiers with Dataset 3.

**Figure 12 ijerph-19-11193-f012:**
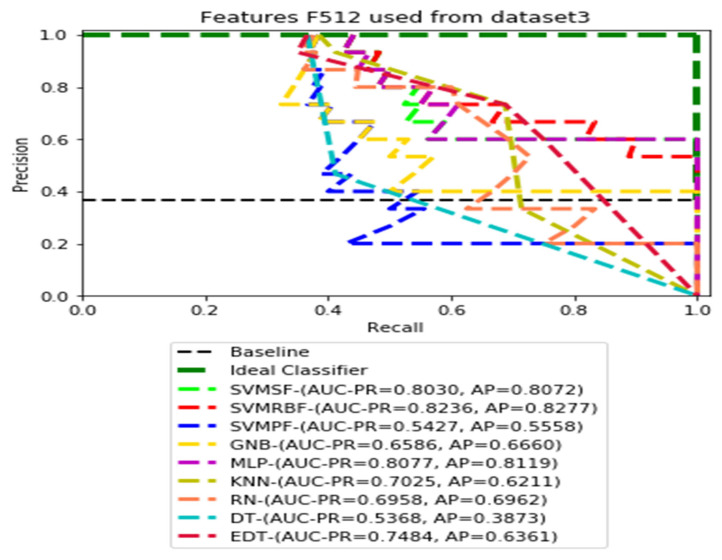
PR curves using feature set F512 as the input of nine classifiers with Dataset 3.

**Figure 13 ijerph-19-11193-f013:**
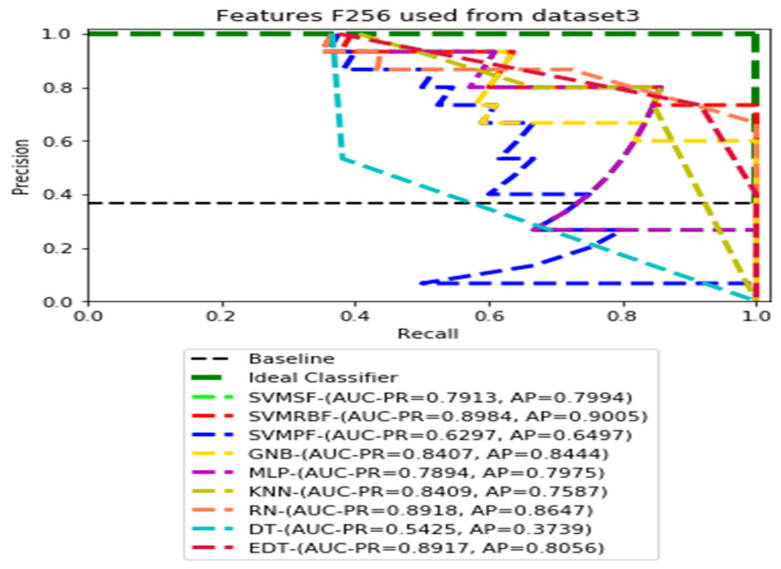
PR curves using feature set F256 as the input of nine classifiers with Dataset 3.

**Figure 14 ijerph-19-11193-f014:**
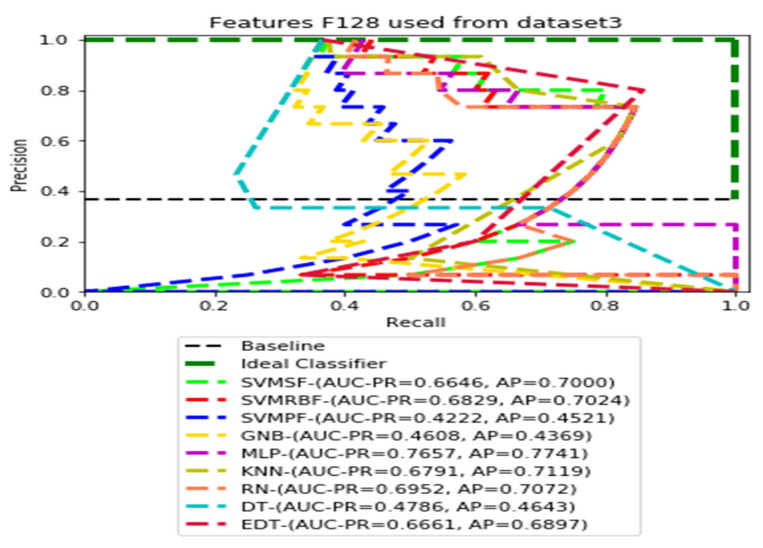
PR curves using feature set F128 as the input of nine classifiers with Dataset 3.

**Figure 15 ijerph-19-11193-f015:**
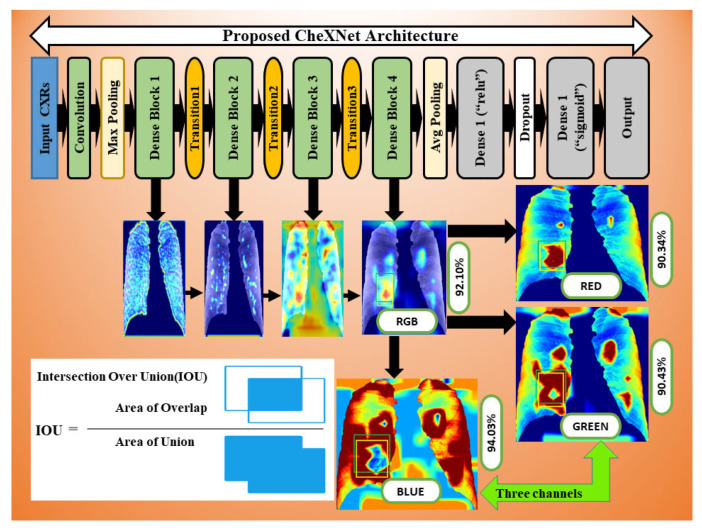
Workflow as the Grad-CAM constructed from four convolutional block CAMs using CheXNet-121.

**Figure 16 ijerph-19-11193-f016:**
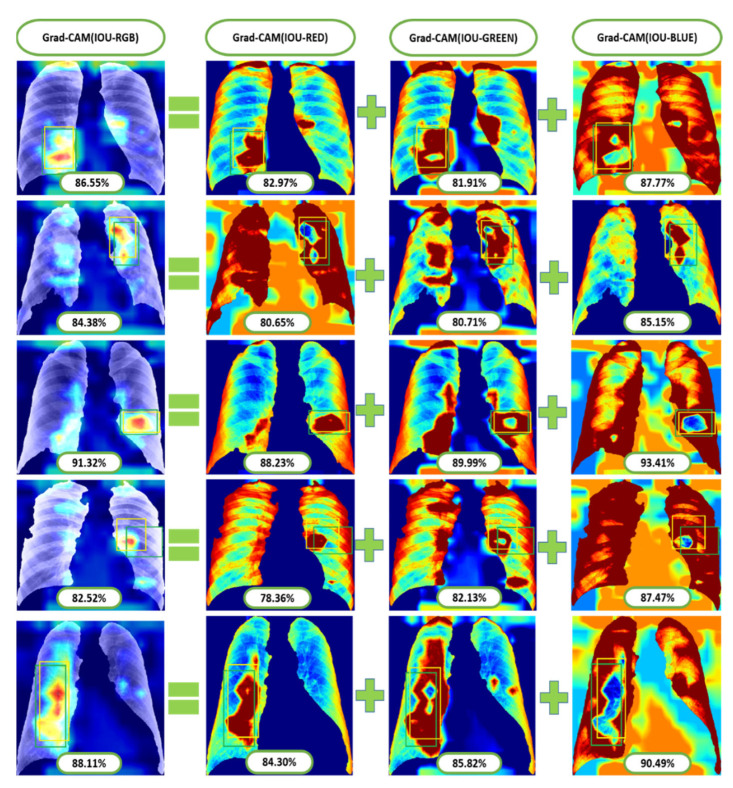
Some examples of Grad-CAM with IOU calculated from test positive samples of the most conspicuous suspect areas.

**Table 1 ijerph-19-11193-t001:** Ensemble results of proposed classifiers among the four feature sets though three datasets (Dataset 1, Dataset 2 and Dataset 3).

MVOT on Feature Sets	Recall (%)	Specificity (%)	Precision (%)	FPR (%)	FNR (%)	F1-Score (%)	Accuracy (%)	MCC (%)
Ensemble-F1024-Dataset 1	92.31	86.67	92.31	13.33	7.69	92.31	90.24	78.97
Ensemble-F1024-Dataset 2	95.83	82.35	88.46	17.65	4.17	92.00	90.24	79.97
Ensemble-F1024-Dataset 3	88.89	85.71	92.31	14.29	11.11	90.57	87.80	73.45
Ensemble-F512-Dataset 1	95.83	82.35	88.46	17.65	4.17	92.00	90.24	79.97
Ensemble-F512-Dataset 2	92.31	86.67	92.31	13.33	7.69	92.31	90.24	78.97
Ensemble-F512-Dataset 3	83.87	100.00	100.00	00.00	16.13	91.23	87.80	74.78
Ensemble-F256-Dataset 1	96.00	87.50	92.31	12.50	4.00	94.12	92.68	84.56
Ensemble-F256-Dataset 2	95.83	82.35	88.46	17.65	4.17	92.00	90.24	79.97
Ensemble-F256-Dataset 3	83.33	90.91	96.15	09.09	16.67	89.29	85.37	68.29
Ensemble-F128-Dataset 1	100.00	83.33	88.46	16.67	00.00	93.88	92.68	85.86
Ensemble-F128-Dataset 2	92.31	86.67	92.31	13.33	7.69	92.31	90.24	78.97
Ensemble-F128-Dataset 3	85.71	84.62	92.31	15.38	14.29	88.89	85.37	67.94

**Table 2 ijerph-19-11193-t002:** Comparison of IOU and AP though three different datasets.

Measurements			Dataset 1	Dataset 2	Dataset 3
**Average-IOU (%)**			89.32	87.85	85.31
**Maximum average-precision (AP) (%)**	**Nine classifiers**	**SVMSF**	91.47	89.82	80.72
		**SVMRBF**	90.41	90.45	90.05
		**SVMPF**	91.38	91.40	78.09
		**GNB**	89.22	90.88	84.44
		**MLP**	90.90	92.34	81.19
		**KNN**	89.87	84.48	75.87
		**RN**	94.90	91.22	86.47
		**DT**	68.63	59.81	46.43
		**EDT**	79.99	83.98	80.56
**Maximum AP-ensemble learning (%)**			92.31	92.31	100.00

## Data Availability

Not applicable.

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
