# Peer review of "Detection and Visualisation of Pneumoconiosis Using an Ensemble of Multi-Dimensional Deep Features Learned from Chest X-rays"

_ijerph, 2022, doi:10.3390/ijerph191811193_

Round 1

Reviewer 1 Report (Previous Reviewer 1)

I am satisfied with the authors responses and the revised version of this paper.

Author Response

We much value your participation in this assessment of our manuscript and like to express our gratitude.

Reviewer 2 Report (Previous Reviewer 3)

Authors proposed the CheXNet model that extracts deep features and ensemble ML models trained using those deep features. Research on Pneumoconiosis seems an important and exciting topic. However, the model is not described clearly. 

- CheXNet is the core model of the study, but it is not described clearly. From Figure 2, the current description is not enough. Previously in the first round of review, I couldn't understand the architecture. However, after reading Huang et al. (2017) paper they cited. I now understand what authors used. Probably authors used one of DenseNet architectures (DenseNet-121, DenseNet-169, DenseNet-201, and DenseNet-264). Authors should use the name of the architecture (DenseNet-xxx). Or they need to describe the definition of the Dense Block and Transition layer they used. Otherwise, readers can not understand. I highly recommend reading Huang et al paper again and re-write model archtecture part of CheXNet. 

Author Response

We much value your participation in this assessment of our manuscript and like to express our gratitude.

Reviewer 3 Report (Previous Reviewer 2)

The authors addressed all my comments and concerns carefully. The paper stands for acceptance now. 

Author Response

We much value your participation in this assessment of our manuscript and like to express our gratitude.

Round 2

Reviewer 2 Report (Previous Reviewer 3)

The authors did a nice job in the revision, and the manuscript has been improved. I don't have further comments or suggestions.

This manuscript is a resubmission of an earlier submission. The following is a list of the peer review reports and author responses from that submission.

Round 1

Reviewer 1 Report

General remarks

§  The paper's title is appropriate.

§  The introduction, research background, datasets and methods, results and comments, and conclusion are all well written.

§  Tables and figures were used to clearly illustrate the results and findings.

§  The authors' exceptional and meritorious contribution to the publication and previous research activities is much appreciated.

§  Experimental data support the uniqueness of the proposed technique. There is great knowledge of recent related work, citations, and references.

§  The research has a significant impact on science and is useful to doctors and other healthcare professionals.

Specific comments

§  The author(s) should rewrite the abstract to properly incorporate the results in terms of the proposed model evaluation performance.

§  To verify that the referencing style adheres to the journal's requirements.

§  The referencing style must be same throughout.

Reviewer 2 Report

The authors presented a better work entitled, Detection and visualisation of pneumoconiosis using an ensemble of multi-dimensional deep features learned from chest x-rays. I believe this work requires minor modifications before acceptance. Authors may consider the following points,

1. A thorough proofread is required

2. A few of the figures resolutions can be increased

3. Results sections may require more elaborations for the readers

Reviewer 3 Report

Authors proposed the CheXNet model that extracts deep features and ensemble ML models trained using those deep features. Research on Pneumoconiosis seems an important and exciting topic. However, the model has several shortcomings.

- CheXNet is the core model of the study, but it is not described clearly. How many CNN layers are used in CheXNet? What are filter sizes and kernel sizes? From Fig 2. it seems there is only one CNN layer is used CheXNet architecture. Isn't it too few? From the ImageNet challenge, winning teams of the year increased the number of CNN layers yearly. (2010,2011: shallow, 2012,2013: 8 layers, 2014: 19 layers, 2015:152 layers). I agree that many CNN layers would result in overfitting with only a few hundred data. However, the use of only one CNN layer seems too few. 

- Authors mentioned transfer learning and argued that they are transferring deep features(F1024, F512, F256, F128). But I think there is a little advantage to the transfer learning part because the dataset only contains a few hundred CXR images. In the image domain, people use pre-trained networks trained on ImageNet data which have about 14 million images. I think researchers should use larger datasets such as NIH CXR datasets for their transfer learning. For training deep features for their transfer learning.